# Cocaine-Induced Reinstatement of Cocaine Seeking Provokes Changes in the Endocannabinoid and *N*-Acylethanolamine Levels in Rat Brain Structures

**DOI:** 10.3390/molecules24061125

**Published:** 2019-03-21

**Authors:** Beata Bystrowska, Małgorzata Frankowska, Irena Smaga, Ewa Niedzielska-Andres, Lucyna Pomierny-Chamioło, Małgorzata Filip

**Affiliations:** 1Department of Toxicology, Collegium Medicum, Jagiellonian University, Medyczna 9, PL 30-688 Kraków, Poland; ewa.niedzielska@uj.edu.pl (E.N.-A.); lucyna.pomierny-chamiolo@uj.edu.pl (L.P.-C.); 2Department of Drug Addiction Pharmacology, Institute of Pharmacology, Polish Academy of Sciences, Smętna 12, PL 31-343 Kraków, Poland; frankow@if-pan.krakow.pl (M.F.); irena.smaga@interia.pl (I.S.); malgorzata.f3@wp.pl (M.F.); 3Department of Internal Medicine, Jagiellonian University Medical College, Skawińska 8, PL 31-066 Kraków, Poland

**Keywords:** cocaine reinstatement, cannabinoid receptors, endocannabinoids, immunohistochemistry, LC-MS/MS method, *N*-acylethanolamines, relapse

## Abstract

There is strong support for the role of the endocannabinoid system and the noncannabinoid lipid signaling molecules, *N*-acylethanolamines (NAEs), in cocaine reward and withdrawal. In the latest study, we investigated the changes in the levels of the above molecules and expression of cannabinoid receptors (CB1 and CB2) in several brain regions during cocaine-induced reinstatement in rats. By using intravenous cocaine self-administration and extinction procedures linked with yoked triad controls, we found that a priming dose of cocaine (10 mg/kg, i.p.) evoked an increase of the anadamide (AEA) level in the hippocampus and prefrontal cortex only in animals that had previously self-administered cocaine. In the same animals, the level of 2-arachidonoylglycerol (2-AG) increased in the hippocampus and nucleus accumbens. Moreover, the drug-induced relapse resulted in a potent increase in NAEs levels in the cortical areas and striatum and, at the same time, a decrease in the tissue levels of oleoylethanolamide (OEA) and palmitoylethanolamide (PEA) was noted in the nucleus accumbens, cerebellum, and/or hippocampus. At the level of cannabinoid receptors, a priming dose of cocaine evoked either upregulation of the CB1 and CB2 receptors in the prefrontal cortex and lateral septal nuclei or downregulation of the CB1 receptors in the ventral tegmental area. In the medial globus pallidus we observed the upregulation of the CB2 receptor only after yoked chronic cocaine treatment. Our findings support that in the rat brain, the endocannabinoid system and NAEs are involved in cocaine induced-reinstatement where these molecules changed in a region-specific manner and may represent brain molecular signatures for the development of new treatments for cocaine addiction.

## 1. Introduction

Cocaine use disorder continues to remain a major world health issue. According to the European Monitoring Centre for Drugs and Drug Addiction (EMCDDA) cocaine is the most commonly used illegal stimulant used in Europe [1] and a relapse in taking it belongs to the more difficult challenges undertaken during the treatment for both the patient and the physician. Moreover, the pathophysiology of this brain disorder still remains unclear.

In recent years, the changes in the lipid signaling molecules such as endocannabinoids and *N*-acetylethyloamines (NAEs) in cocaine reward and the drug’s withdrawal have been demonstrated [2,3,4,5,6]. In fact, self-administration of cocaine induced both a potent decrease of the anandamide (AEA) levels in the cerebellum and in the 2-arachidonoylglycerol (2-AG) levels in the hippocampus and striatum with fewer cannabinoid CB1 receptors, while increased 2-AG levels were observed in the frontal cortex and cerebellum and were was linked with cocaine reward in rats [2,7]. A cocaine-free period with extinction training resulted in a potent reduction of the endocannabinoid levels in the limbic and subcortical areas with the cannabinoid CB2 receptor downregulation [7].

Apart from endocannabinoids, which are direct agonists of CB receptors, there are also other lipid-based mediators—NAEs (oleoylethanolamide (OEA) and palmitoylethanolamide (PEA))—and they act indirectly on the cannabinoid receptors. NAEs are the endogenous ligands of the nuclear receptor peroxisome proliferator-activated receptors (PPAR-α) and the transient receptor potential cation channel subfamily V member 1 (TRPV1) [8]. PEA and OEA levels could strengthen the effect of AEA on the cannabinoid receptor or TRPV1 (“entourage effect”) [9] and consequently potentiate the effect of endocannabinoids [10]. Contingent cocaine administration upregulated the NAEs (oleoylethanolamide (OEA) and palmitoylethanolamide (PEA)) levels in the limbic areas [2], while the levels of these molecules were also seen to increase during extinction in the prefrontal cortex and the hippocampus only during the rats’ self-administered cocaine phase [2,7].

In behavioral preclinical studies, the evidence for engagement of the endocannabinoid system on drug-seeking behavior was demonstrated using specific CB1 receptor antagonists that attenuated cocaine-primed and cue-induced reinstatement of cocaine-seeking behavior in rats [11,12,13].

To further examine the involvement of endocannabinoids and NAEs in cocaine addiction, this study attempted to reveal the role of AEA, 2-AG, OEA, and PEA in cocaine relapse. Through using the drug-primed procedure with a yoked triad control with rats, we studied the molecule tissue levels with using liquid chromatography with a tandem mass spectrometry (LC-MS/MS) method. Finally, we examined the cannabinoid CB1 and CB2 receptor brain expression following cocaine reinstatement. We selected several rat brain structures related to substance use disorders including prefrontal cortex (executive control), amygdala and hippocampus (memory and learning), dorsal striatum (habit forming learning), nucleus accumbens (reward processing), ventral tegmental area (reward-related and goal-directed behaviors), and cerebellum (nonmotor function, intertwined with brain reward, motivational drive, saliency, inhibitory control, and insight processes) [14,15,16].

The present study on the role of the endocannabinoid system in the cocaine dependence mechanism may contribute to the many missing questions about cocaine’s interaction of the endocannabinoid system, and may help to determine whether the endocannabinoid system is the target point for new addiction treatment drugs in the future.

## 2. Results

### 2.1. Behavioral Studies

After 14 self-administration sessions, the animals showed stable lever-pressing rates during the last 3 self-administration days with less than a 10% difference in their daily intake of cocaine (Figure 1). The mean number of cocaine infusions per day during the last 3 self-administration days varied from 26 to 29 while during 14 experimental sessions the animals received 164.9 ± 9.2 mg/rat of cocaine.

Figure 1 shows the behavioral responses of the rats that underwent cocaine self-administration, extinction training, and cocaine-induced reinstatement of seeking behavior. A two-way ANOVA for repeated measures for animals previously self-administered cocaine for tissue level of endocannabinoids and NAEs and expression of cannabinoid receptor experiments showing the significant effect of lever × session interaction (F(24,336) = 10.72; *p* < 0.001 and F(24,336) = 9.47; *p* < 0.001, respectively). The post hoc analyses revealed a greater frequency of presses on the “active” lever than on the “inactive” lever from the 1st or 2nd cocaine self-administration session till the 1st or 2nd extinction day. During extinction training, neither drug nor drug-paired stimuli were given in response to lever pressing, resulting in a gradual decrease in “active” lever presses (*p* < 0.001). Moreover, an intraperitoneal injection of cocaine induced a significant increase of “active” lever presses during the reinstatement of drug seeking in rats that previously self-administered cocaine (*p* < 0.01). There was no alteration in inactive lever presses after exposure to priming.

In the yoked cocaine and yoked saline groups, the difference between pressing the “active” versus the “inactive” lever failed to reach significance both in the experiment used to evaluate the endocannabinoid and NAEs tissue level and in the expression of cannabinoid receptors (a two-way ANOVA for repeated measures: Yoked cocaine + i.p. cocaine: F(24,336) = 0.83; *p* = 0.69 and F(24,288) = 1.27; *p* = 0.18; Yoked saline + i.p. cocaine: F(24,336) = 0.59; *p* = 0.79 and F(24,288) = 0.31; *p* = 0.99; Yoked saline + i.p. saline: F(24,336) = 0.45; *p* = 0.83 and F(24,288) = 0.31; *p* = 0.99, respectively).

### 2.2. Tissue Level of Endocannabinoid and NAEs in Rat Brain Structures

In the yoked saline + i.p. saline group, the AEA levels ranged from 11.37 to 16.33 ng/g, with the highest concentration observed in the nucleus accumbens and the lowest in the frontal cortex. As shown in Figure 2, reinstatement of seeking behavior in the animals which had undergone a prior procedure of cocaine self-administration resulted in a changes in the AEA levels in all of the analyzed brain structures: the cerebellum (F(3,30) = 12.52; *p* < 0.001), nucleus accumbens (F(3,28) = 23.41; *p* < 0.001), hippocampus (F(3,30) = 6.43; *p* < 0.01), dorsal striatum (F(3,29) = 4.88; *p* < 0.05), frontal cortex (F(3,28) = 6.70; *p* < 0.05), and prefrontal cortex (F(3,28) = 12.46; *p* < 0.001). A significant decrease in the AEA level was observed in the cerebellum and nucleus accumbens in the animals which self-administered (at least *p* < 0.01) or passively received (*p* < 0.001) cocaine in comparison to the yoked saline animals. Instead, in the prefrontal cortex (*p* < 0.01) and hippocampus (*p* < 0.05) an increase was reported only in cocaine self-administered rats. Moreover, the animals that had previously self-administered cocaine showed a reduction of the AEA level in the dorsal striatum and frontal cortex when compared to the yoked saline + i.p. cocaine and yoked cocaine + i.p. cocaine rats which, during restatement of seeking-behavior, received an intraperitoneal injection of cocaine (10 mg/kg).

The concentration of 2-AG in the control (yoked saline + i.p. saline) group ranged from 3.21 to 5.12 μg/g, with the highest concentration in the dorsal striatum and the lowest in the nucleus accumbens. Cocaine-induced reinstatement influenced the 2-AG levels in the following structures: the frontal cortex (F(3,28) = 11.27; *p* < 0.001), hippocampus (F(3,29) = 33.38; *p* < 0.001), dorsal striatum (F(3,30) = 26.22; *p* < 0.001), nucleus accumbens (F(3,29) = 3.65; *p* < 0.05), and cerebellum (F(3,28) = 6.23; *p* < 0.01), but not in the prefrontal cortex (F(3,29) = 2.74). The post hoc analyses revealed a significant decrease in the 2-AG level in the frontal cortex (*p* < 0.001) and cerebellum (*p* < 0.05), whilst in the hippocampus (*p* < 0.001) and in the nucleus accumbens (*p* < 0.05), an increase was reported in rats that had self-administered cocaine in comparison to the corresponding yoked saline + i.p. saline group. In the animals passively administered cocaine, a significant enhancement in the 2-AG level in the dorsal striatum (*p* < 0.001) and a significant reduction in the cerebellum (*p* < 0.05) was observed (Figure 3).

In the yoked saline + i.p. saline group, the OEA levels ranged from 21.78 to 58.78 ng/g, with the highest concentration in the nucleus accumbens and the lowest in the prefrontal cortex. Cocaine treatment during reinstatement of seeking behavior resulted in a change in the OEA level in the all studied structures: the prefrontal cortex (F(3,30) = 24.36; *p* < 0.001), frontal cortex (F(3,30) = 3.82; *p* < 0.05), hippocampus (F(3,30) = 29.15; *p* < 0.001), dorsal striatum (F(3,29) = 53.84; *p* < 0.001), nucleus accumbens (F(3,30) = 69.02; *p* < 0.001), and cerebellum (F(3,29) = 20.80; *p* < 0.001). In the rats that had self-administered cocaine and those given yoked cocaine injections, a significant increase in the OEA levels was observed in the prefrontal cortex (*p* < 0.001) and dorsal striatum (*p* < 0.001) when compared to the yoked saline + i.p. saline rats. On the other hand, both groups previously administered cocaine, actively or passively, demonstrated a significant reduction of the level of OEA in the hippocampus (*p* < 0.001) and cerebellum (*p* < 0.001). Moreover, the animals previously self-administered cocaine displayed a reduction of the OEA level in the nucleus accumbens (*p* < 0.001) in comparison to all of the yoked groups. The Dunnett’s Multiple Comparison test also revealed a significant decrease of the OEA level in the hippocampus of the rats which had previously received saline and were exposed to noncontingent reinforcements (*p* < 0.001) when compared to the corresponding yoked saline + i.p. saline group (Figure 4).

The concentration of PEA ranged from 30.90 to 83.60 ng/g in the control (yoked saline + i.p. saline) group, with the highest concentration in the nucleus accumbens and the lowest in the prefrontal cortex. Cocaine administration induced changes in the PEA levels in all of the structures (the prefrontal cortex (F(3,30) = 231.20; *p* < 0.001), frontal cortex (F(3,30) = 59.06; *p* < 0.001), hippocampus (F(3,29) = 6.59; *p* < 0.001), dorsal striatum (F(3,29) = 68.15; *p* < 0.001), nucleus accumbens (F(3,30) = 75.41; *p* < 0.001), and cerebellum (F(3,29) = 81.28; *p* < 0.001)). During cocaine-induced reinstatement in rats previously self-administered cocaine, an increase in the PEA levels was observed in the prefrontal cortex (*p* < 0.001), the hippocampus (*p* < 0.05), and dorsal striatum (*p* < 0.001), while a decrease was noted in the nucleus accumbens (*p* < 0.001) and cerebellum (*p* < 0.001). In rats passively administered cocaine, increases in the PEA levels in the frontal cortex (*p* < 0.001) and dorsal striatum (*p* < 0.001) and decreases in the cerebellum (*p* < 0.001) were reported (Figure 5).

### 2.3. Expression of the CB1 and CB2 Receptor Protein

In the cocaine-induced reinstatement phase, we observed a decrease in the expression of CB1 receptor protein in the ventral tegmental area (F(3,24) = 4.95; *p* < 0.01) in both of the studied groups and in the control group with a cocaine i.p. dose. Furthermore, significant changes were observed in the lateral septal nuclei (F(3,24) = 8.54; *p* < 0.001) and prefrontal cortex (F(2,14) = 8.46; *p* < 0.01), and an increase of the expression of CB1 receptors was observed in the group of rats that had actively self-administered cocaine (at least *p* < 0.05). Interestingly, significant changes were demonstrated in the dorsal striatum (F(3,24) = 4.93; *p* < 0.05), and a decrease in the CB1 receptors level was observed in the yoked cocaine + i.p. cocaine group. A one-way ANOVA did not show changes in CB1 receptor expression in the basolateral and basomedial amygdala (F(3,24) = 1.06), nucleus accumbens core (F(3,24) = 2.43) and shell (F(3,24) = 0.46), medial globus pallidus (F(3,24) = 1.69), hippocampus (F(3,24) = 1.23), and substantia nigra (F(3,24) = 2.72). The results of the analysis are shown in Figure 6 (top panel).

In the CB2 receptor expression, we noticed statistically significant changes in the lateral septal nuclei (F(3,24) = 9.85; *p* < 0.001), and the post hoc tests demonstrated an increase in the receptor level in chronic and acute cocaine-treated rats (*p* < 0.001). Moreover, significant changes were noted in the prefrontal cortex (F(3,24) = 11.39; *p* < 0.001), and an increase in the receptors was noticed in active cocaine self-administered (*p* < 0.001) and yoked saline + i.p. cocaine (*p* < 0.05) animals exposed to cocaine intraperitoneal injection during reinstatement. Furthermore, significant changes were demonstrated in medial globus pallidus (F(3,24) = 6.53; *p* < 0.01), while an increase was observed only in yoked cocaine + i.p. cocaine rats (*p* < 0.01). Also in the dorsal striatum a significant changes in the CB2 receptor expression were indicated (one-way ANOVA (F(3,24) = 3.57; *p* < 0.05); a decrease in the receptors level in the passive cocaine group was noted in comparison to the other experimental groups. In the nucleus accumbens core (F(3,24) = 1.14), nucleus accumbens shell (F(3,24) = 0.29), basolateral and basomedial amygdala (F(3,24) = 0.80), the substantia nigra (F(3,24) = 2.58), hippocampus (F(3,24) = 1.60), and ventral tegmental area (F(3,24) = 1.77), the level of CB2 receptors was unchanged. The results of analysis are shown in Figure 6 (bottom panel).

Table 1 shows the observed changes in tissue levels of eCS and NAEs and changes in the expression of cannabinoid receptors (CB1 and CB2) in the rat brain structures.

## 3. Discussion

The present study demonstrates the alterations in the level of endocannabinoids and cannabinoid CB1 and CB2 receptors in the rat brain structures following a cocaine-induced relapse. At the same time, we also found cocaine-induced changes in OEA and PEA concentrations after the cocaine challenge dose in the animals with a previous motivational and experimental delivery of cocaine. This paper follows our previous study where we reported several changes in the concentration of AEA and 2-AG as well as OEA and PEA linked to cocaine reward or to cocaine withdrawal [2].

In the current study, a cocaine priming dose resulted in a significant increase of the AEA level in the hippocampus and prefrontal cortex only in animals with a previous motivational intake of cocaine. Since the changes were not present in yoked saline + i.p. cocaine or cocaine + i.p. cocaine controls (present study), during cocaine self-administration and in the cocaine-free period with extinction training [2], it means the specific effect of a cocaine-induced relapse is attributed to the limbic AEA levels. The reductions of the AEA levels in the frontal cortex and striatum linked to previous cocaine self-administration and in the nucleus accumbens and cerebellum due to repeated cocaine injections did not reflect cocaine-induced reinstatement but, instead, rather persistent adaptations after chronic cocaine exposure [2].

With regards to the second endocannabinoid studied, the 2-AG levels also increased in the hippocampus and nucleus accumbens while they decreased in the frontal cortex only after cocaine priming in rats previously self-administered cocaine. The recent literature data confirms the involvement of the endocannabinoid molecules (especially 2-AG) in the hippocampus in the (i) modulation of the dopaminergic system [17,18] and in (ii) cocaine-induced reinstatement [19]. In fact, we reported that systemic injections with blockers of fatty acid amide hydrolase (FAAH), the main enzyme responsible for terminating the actions of the endocannabinoid anandamide, was able to reduce a cocaine-primed relapse in rats [20]. The protective effects of FAAH inhibitions were also noted for cocaine seizures and cell death in the hippocampus in a model of the drug’s intoxication in mice [21]. Furthermore, previous research on the endocannabinoids and stress responses indicated a significant role of 2-AG in regulating stress behaviors [22]. In chronic stress, the 2-AG level increased [23], and the rise in the striatal 2-AG level in may indicate adaptation to a stressful situation in the rats that had had a passive cocaine intake.

Parallel immunohistochemical analyses revealed that cocaine priming enhanced the expression of the CB1 and CB2 receptors in the prefrontal cortex and lateral septal nuclei. The upregulation of the CB1 receptors observed in the prefrontal cortex and the lateral septal nuclei was present during cocaine self-administering rats and should be linked to the drug rewarding properties [7]. The lateral septum is the middle of the relay for connections from the CA3 hippocampus to the ventral tegmental area, which makes it possible to link the reward signals with the context in which they occur [4]. An increase in the expression of CB1 and CB2 receptors observed in the lateral septal nuclei may also be the result of conditional stimuli accompanying the experiment (sound and light signal associated with the dose of the drug) [24]. The present neurochemical data supports our previous behavioral findings in which we reported that tonic activation of CB1 is involved in cocaine-induced reinstatement of cocaine-seeking behavior [20]. Interestingly, we also observed a decrease in the CB1 receptor expression in the ventral tegmental area seen in all of the studied rat groups, including the animals treated with acute cocaine. Reduced levels of CB1 receptors in the ventral tegmental area in all of the experimental groups suggest a pharmacological effect of cocaine-priming. In the ventral tegmental area, CB1 receptors are expressed on GABA neurons and their reduction observed in this paper suggests a pharmacological effect of cocaine-priming leading to the rise of the dopamine level and increased sensitivity to the drug effects [3,25,26]. Upregulation of the CB2 receptors seems to be related to the last cocaine dose, because an increase in the CB2 receptor expression was also presented in the control group with a single i.p. cocaine dose.

The levels of endocannabinoids reflect their concentration at the time of tissue collection for analysis (the neurotransmitter presence in the synaptic space and in presynaptic neurons), which may not necessarily correlate with the expression of cannabinoid receptors. Similar results were reported during chronic stress, where the 2-AG level was increased [23], while the expression of CB1 receptor did not change in any brain regions [27].

Recently, Luchicchi et al. (2010) demonstrated that NAEs play a role in behavioral and neurochemical effects related to cocaine addiction [28]. The finding from this study showed that NAEs are involved in the relapse of cocaine-seeking behavior. OEA and PEA are endogenous lipid transmitters with similar properties to AEA and 2-AG, although they did not act on cannabinoid receptors. They act through the nuclear proliferator-activated receptor alpha (PPARα), the transient receptor potential vanilloid 1 (TRPV1), and the orphan G-protein-coupled receptors GPR55 and GPR119. NAEs participate in the regulation of many physiological processes and they have neuroprotective, antinociceptive, and anti-inflammatory effects [29]. In this paper, we found that a cocaine-induced relapse resulted in a potent increase in OEA levels in the prefrontal cortex and striatum and in PEA in the prefrontal cortex, hippocampus and dorsal striatum. At the same time, there were potent reductions in the NAEs level in the rat cerebellum. The above changes were noted for chronic cocaine experienced rats, while a decrease in the tissue levels of OEA and PEA observed in the nucleus accumbens was linked with cocaine relapse in rats. Decreased levels of OEA and PEA in nucleus accumbens after the cocaine challenge only in rats self-administering cocaine suggest an impaired control of the release of neurotransmitters during reinstatement, and correlate with the previously reported the participation of the TRPV1 receptor [13,20,28]. Recently, Zambrana-Infantes et al. (2018) demonstrated that repeated administration of PEA can block behavioral cocaine sensitization in the mechanism, probably independent of the PPAR-α receptor, but rather as a result of the anti-inflammatory properties of PEA [29]. The decrease in PEA and OEA in the nucleus accumbens observed in our study may confirm the increased cocaine sensitization as a result of the habit forming, since the same changes have not been observed in cocaine-experienced animals following repeated passive (yoked) treatment. It is also worth noting that acute administration of OEA reduced cocaine locomotion or reward but did not affect cocaine sensitization, conditioned place preference, and reinstatement to cocaine in PPARα receptor knockout in mice [30]. The decrease in PEA and OEA in nucleus accumbens observed in our study may confirm the increased cocaine sensitization after a cocaine-priming dose as a result of the habit forming, but pharmacological analyses with selective ligands are required.

## 4. Materials and Methods

### 4.1. Animals

Male Wistar rats (280–300 g, N = 70) delivered by a licensed breeder (Charles River, Germany) were housed individually in standard plastic rodent cages in a colony room maintained at 22 ± 2 °C and at 45–65% humidity under a 12-h light–dark cycle (lights on at 06:00). The animals had free access to standard animal food and water during the 7-day habituation period. We used a yoked procedure to assess whether a motivational factor would have an impact. Then, the animals were divided into four groups: active cocaine + i.p. cocaine, yoked cocaine+ i.p. cocaine, yoked saline + i.p. cocaine, and yoked saline + i.p. saline and those used in the cocaine self-administration procedures were maintained on limited water during the initial training sessions (see below). All of the experiments were conducted during the light phase of the light–dark cycle (between 8.00 a.m. and 3.00 p.m.) and the experiments were carried out in accordance with the European Directive 2010/63/EU and approved by the Ethical Committee at the Institute of Pharmacology, Polish Academy of Sciences, Krakow. All of the animals used in the study were experimentally naïve.

### 4.2. Drugs

Cocaine hydrochloride (Sigma-Aldrich, St. Louis, MO, USA), dissolved in sterile 0.9% NaCl and given iv (0.1 mL/infusion) or i.p. (1 mL/kg).

### 4.3. Behavioral Procedures

#### 4.3.1. Cocaine Self-Administration, Extinction Training and Reinstatement of Drug-Seeking Behavior

After a week of habituation to the animal facility, animals were water-deprived for 18 h and subsequently trained to press the lever of standard operant conditioning chambers (Med-Associates, ST Albans City, VT, USA) under an increasing schedule of water reinforcement at a fixed ratio (FR) of 1 to 5. Two days following “lever-press” training under FR5 and free access to water, the rats were chronically implanted with a silastic catheter in the external right jugular vein, as described previously [31]. The catheters were flushed every day with 0.1 mL of saline solution containing heparin (70 U/mL, Biochemie GmbH, Kundl, Austria) and 0.1 mL of solution of cephazolin (10 mg/mL; Biochemie GmbH, Kundl, Austria).

After a 7–8-day recovery period, all of the animals were water deprived for 18 h and trained to lever press to a FR5 schedule of water reinforcement over a 2-h session. Some of the subjects (active cocaine groups) were then given access to cocaine during 2-h daily sessions performed 6 days/week (maintenance) and from that time they were given ad libitum water. The house light was illuminated throughout each session. Each completion of five presses on the “active” lever complex (the FR5 schedule) resulted in a 5-s infusion of cocaine (0.5 mg/kg per 0.1 mL) and a 5-s presentation of a stimulus complex (activation of the white stimulus light directly above the “active” lever and the tone generator, 2000 Hz; 15 dB above ambient noise levels). Following each injection, there was a 20-s time-out period during which the responses were recorded but had no programmed consequences. Those on the “inactive” lever did not result in cocaine delivery. Acquisition of the conditioned operant response lasted a minimum of 10 days until the subjects met the following criteria: active lever presses with an average of 3 consecutive days and a standard deviation within those 3 days of <10% of the average; this selected criterion was based on our prior experiments [32]. The animals trained to self-administer cocaine (0.5 mg/kg/infusion) (which met the acquisition criterion) were used in the extinction study and next in the reinstatement test. During extinction training, the subjects had 2-h daily training sessions with no cocaine delivery (saline was substituted for cocaine) and no presentation of the conditioned stimulus. Once they reached the extinction criteria (a minimum of 10 extinction days and with the responses to the active lever below 10% of the level observed during maintenance over at least 3 consecutive days), on the 10th day of extinction, the animals were tested for the response reinstatement induced by a noncontingent presentation of a self-administered reinforcements (10 mg/kg cocaine i.p.). During the reinstatement tests (2-h sessions), active lever presses on the FR5 schedule resulted in intravenous injection of saline. All of the animals were sacrificed after a 2-h experimental session.

#### 4.3.2. Yoked Self-Administration Procedure

The rats were tested simultaneously in groups of three with two rats serving as ‘yoked’ controls that received an injection of saline or cocaine which was not contingent on responding, and each time a response-contingent injection of 0.5 mg/kg cocaine was self-administered by the paired rat. Unlike the self-administering rats, the lever pressing by the yoked rats was recorded but had no programmed consequences [33]. The yoked saline + i.p. saline, saline + i.p. cocaine, and cocaine + i.p. cocaine animals were sacrificed at the same time as the corresponding self-administered cocaine groups of rats after a 2-h experimental session of reinstatement drug seeking. The plan of the experiment is presented in Figure 7.

### 4.4. Neurochemical Analysis

#### 4.4.1. LCMS/MS Analysis

##### Brain Structures Isolation and Tissue Preparation

The rats were sacrificed through decapitation. Selected brain structures (i.e., the prefrontal cortex, frontal cortex, hippocampus, dorsal striatum, nucleus accumbens, and cerebellum; Figure 8 panel A) were isolated, immediately frozen on dry ice and stored at −80 °C. the tissues were dissected out according to The Rat Brain Atlas and to the chapter in Neuroproteomics [34,35].

##### Reagents

Standards of AEA, 2-AG, PEA, and OEA were obtained from Tocris (Bristol, United Kingdom), AEA-d_4_, 2-AG-d_5_, PEA-d_4_ and OEA-d_4_ were from Cayman Chemical (Ann Arbor, MI, USA), acetonitrile and chloroform were from Merck (Darmstadt, Germany), while methanol and formic acid were from POCh (Katowice, Poland). All the stock solutions of the standards, excluding 2-AG and 2-AG-d_5_, were prepared in ethanol, whereas 2-AG and 2-AG-d_5_ stock solutions were prepared in acetonitrile (concentration 1 mg/mL). All the stock solutions were stored at −80 °C. Further dilutions were prepared using acetonitrile.

##### Lipid Extraction from Brain Tissue

The extraction was carried out by a method of lipid compounds’ isolation developed by Folch in 1957 with our modifications [36]. The tissues were homogenized in an ice-cold mixture of methanol and chloroform (1:2, *v*/*v*) at the proportion of 10 mg of wet tissue per 150 μL of solvent. Next, 150 μL of the homogenate was mixed with 250 μL of formic acid (pH 3.0; 0.2 M) and 1500 μL of the extraction mixture (methanol: chloroform; 1:2, *v*/*v*). Then, 2 μL of an internal standard (AEA-d4, at 10 μg/mL; 2-AG-d_5_ at 100 μg/mL, and PEA-d_4_, OEA-d_4_ at 5 μg/mL) were added to each sample. The samples were afterwards vortexed for 30 s and centrifuged at 2000 rpm for 10 min. The organic phases were collected and dried under a nitrogen stream at 40 °C. The dry residue was dissolved in 40 μL of acetonitrile and 10 μL of the reconstituted extract was injected into the LC-MS/MS system for a quantitative analysis purpose.

##### Chromatographic Analysis: LC-MS/MS Conditions

The LC-MS Applied Biosystem API 2000 (Perlan Technologies, Warsaw, Poland) instrumentation consisted of the following components: an Agilent series 1100 pump (Santa Clara, CA, USA) in its gradient mode and an autosampler Agilent series 1100 with a 100 µL injection loop.

The chromatographic separation was performed in the gradient mode with a Thermo Scientific BDS HYPERSIL (Waltham, MA, USA) C18 column 100 × 3 mm I.D., 3 µm particle size. The advance column, with a precolumn (10 × 3 mm I.D., 3 µm particle size), was maintained at 40 °C and the mobile phase flow rate was 0.3 mL/min. The mobile phases for gradient elution consisted of A (formic acid (0.02 M) in acetonitrile) and B (formic acid (0.02 M) in water). The gradient started initially at 0% A over 1 min, then increased linearly to 90% by 2 min, where it stayed for the next 2 min; before the solvent A content decreased to 0% over 6 min. Finally, for the last 4 min of the analysis, the column was maintained at 100% B to stabilize the baseline. The sample temperature was maintained at 4 °C in the autosampler prior to analysis.

Quantification of the analytes was performed using the tandem electrospray MS in a positive mode (ESI). The data acquisition and processing were accomplished using the Applied Biosystems Analyst version 1.4.2 software. Endocannabinoid and NAE concentrations in the samples were calculated with the calibration curve prepared using the brain tissues samples collected from naive rats (rats not previously subjected to experimentation, without a particular experimental situation and without a particular drug/antigen used before) to eliminate the matrix effect. The brain tissues samples were spiked with AEA, PEA, and OEA to obtain the concentration of 0.2; 2; 20; 50; 100; 200 ng/g and with 2-AG to obtain the concentration of 0.8; 2; 10; 20; 50; 100 µg/g. AEA-d_4_, 2-AG-d_5,_ PEA-d_4_ and OEA-d_4_ were used as the internal standard. The procedure described for the sample preparation was followed for sample analysis.

The endocannabinoid levels were determined from the calibration curve by means of linear regression applied to the response ratios (peak area for specific analyte/peak area for the internal standard) as a function of the corresponding endocannabinoid/NAE concentration. To ensure the method was sufficiently thorough, analytical assessments for selectivity, linearity, precision, accuracy, recovery, and stability were performed [36].

#### 4.4.2. Immunohistochemistry

The rats were sacrificed immediately after the experimental session with pentobarbital and perfused intracardially with a solution of 4% paraformaldehyde (Sigma Aldrich, St. Louis, MO, USA) in 100 mM phosphate buffer, pH 7.4 (PBS; 81 mM Na_2_HPO_4_, 19 mM NaH_2_PO_4_, 27 mM KCl, 154 mM NaCl; POCH, Gliwice, Poland). The brains were excised and post-fixed in the same fixative for 12 h. The tissue was permeated with in 10% *w*/*v* sucrose (Sigma Aldrich, St. Louis, MO, USA) for 7 days and removed in 30% *w*/*v* sucrose in PBS at 4–8 °C for no less than 48 h. The brains were deeply frozen on dry ice and cut into 12 μm coronal sections on a cryostat (Leica Microsystems, Nussloch, Germany) and were kept at −20–−22 °C until processed for immunohistochemistry.

##### Immunohistochemistry Procedure

The rat brain sections were rinsed in 100 mM PBS buffer pH 7.5, and in PBS containing 0.1% Triton X-100 (Sigma Aldrich, St. Louis, MO, USA), for 30 min at room temperature and Odyssey Blocked Buffer (OBB; Li-COR Biosciences, Cambridge, UK) for 1h at room temperature. The tissue sections were incubated overnight at 4 °C with purified primary antibodies in OBB containing 0.1% Tween 20: 1:200 dilution of rabbit monoclonal anti-CB_1_ receptor (Abcam, ab172970), and 1:200 dilution of goat polyclonal anti-CB2 (Santa-Cruz, sc-10076; [37]). Following four washes of 5 min each with PBS containing 0.1% Tween 20, the tissue sections were incubated for 1 h at RT with goat anti-rabbit (IRDye^®^ 680CW, Li-COR Biosciences, Cambridge, UK) and donkey anti-goat (IRDye^®^ 800CW, Li-COR Biosciences, Cambridge, UK) secondary antibodies at a dilution of 1:2000. Fluorescence was detected using the Odyssey^®^ Infrared Imaging System (21 μm resolution, 1 mm offset with highest quality) (Li-COR, Lincoln, NE, USA). Channel sensitivity was optimized for each set of stained sections and maintained for that group of samples. The relative location of the slice and identification of brain regions were determined by comparison to images in the Paxinos and Watson Rat Brain Atlas (2007) [7] (Figure 8 panel B).

### 4.5. Data Analysis

Animals that had problems with catheters during the recovery period, not performing the self-administered acquisition and extinction criteria or those showing incorrect perfusion were excluded from the data analysis. All the data were expressed as means (± SEM). Behavioral data were analyzed by a two-way ANOVA for repeated measures and for statistical evolution of these data across the following factors: lever (“active” and “inactive”) and sessions where self-administration, extinction training, and reinstatement of seeking behavior were used. Significant group differences according to an ANOVA were analyzed by the post hoc Newman–Keuls test. Neurochemical data were analyzed by one-way ANOVA for different treatment groups (Yoked saline + i.p. saline, Yoked saline + i.p. cocaine, Yoked cocaine + i.p. cocaine, and Active cocaine + i.p. cocaine) and appropriate, post hoc comparisons were conducted by a Dunnett’s Multiple Comparison test. A *p* value < 0.05 was considered as statistically significant.

## 5. Conclusions

We found that endocannabinoids (AEA and 2-AG) and endogenous noncannabinoid lipid signaling molecules (PEA, OEA) are engaged in the reinstatement of cocaine-seeking behavior in rats. Our results support behavioral analyses with selective cannabinoid, TRPV1 or PPAR-α receptor ligands to control cocaine addiction. The present findings extend our previous observations that the endocannabinoid and NAE molecules and CB1 receptors influence cocaine self-administration.

## Figures and Tables

**Figure 1 molecules-24-01125-f001:**
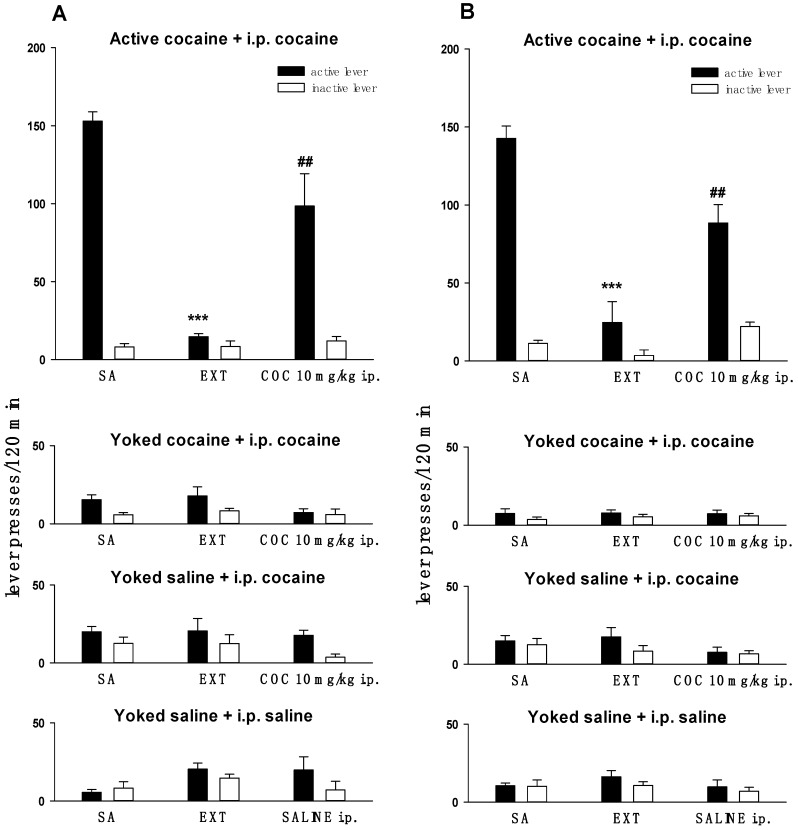
The number (mean ± SEM) of active (black bars) and inactive (white bars) lever presses during the last three self-administration (SA) and extinction (EXT) sessions in rats trained to self-administered cocaine (0.5 mg/kg/infusion) and forced to cocaine priming (COC 10 mg/kg i.p.) during the reinstatement test in active cocaine + i.p. cocaine, yoked cocaine + i.p. cocaine, yoked saline + i.p. cocaine and yoked saline + i.p. saline was shown. Panel **A**—experiment to evaluate the endocannabinoid and NAEs tissue level; panel **B**—experiment to evaluate the expression of cannabinoid receptors. N = 7–8 rats/group. *** *p* < 0.001 vs. active lever SA; ^##^
*p* < 0.01 vs. active lever EXT, (Newman–Kelus test).

**Figure 2 molecules-24-01125-f002:**
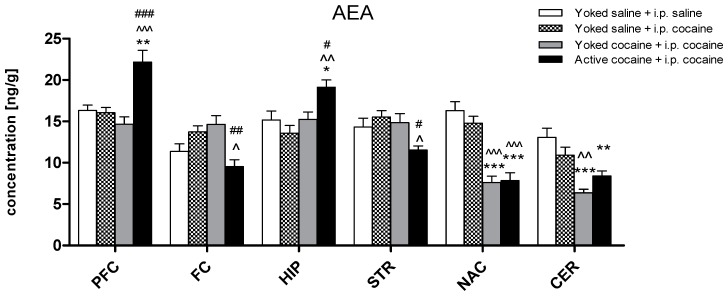
Levels of anadamide (AEA) in brain structures in rats previously self-administered cocaine (14 days) (Active cocaine) or which passively received cocaine (Yoked cocaine) or saline (Yoked saline) and underwent extinction training (10 days) and drug-induced reinstatement (10 mg/kg cocaine i.p. or saline i.p.). N = 7–8 rats/group. All data are expressed as the means ± SEM. * *p* < 0.05, ** *p* < 0.01, *** *p* < 0.001 vs. Yoked saline + saline i.p.; ^ *p* < 0.05, ^^ *p* < 0.01, ^^^ *p* < 0.001 vs. Yoked saline + cocaine i.p.; ^#^
*p* < 0.05, ^##^
*p* < 0.01, ^###^
*p* < 0.001 vs. Yoked cocaine + cocaine i.p. PFC—prefrontal cortex; FC—frontal cortex, HIP—hippocampus, STR—dorsal striatum, NAC—nucleus accumbens, CER—cerebellum.

**Figure 3 molecules-24-01125-f003:**
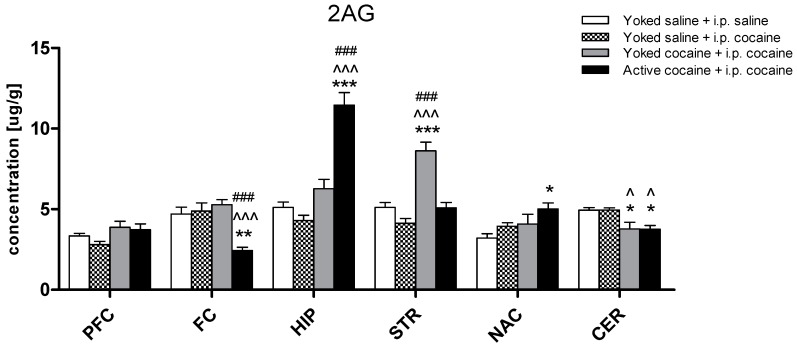
Levels of 2-arachidonoylglycerol (2-AG) in brain structures in rats previously self-administered cocaine (14 days) (Active cocaine) or which passively received cocaine (Yoked cocaine) or saline (Yoked saline) and underwent extinction training (10 days) and drug-induced reinstatement (10 mg/kg cocaine i.p. or saline i.p.). N = 7–8 rats/group. All data are expressed as the means ± SEM. * *p* < 0.05, ** *p* < 0.01, *** *p* < 0.001 vs. Yoked saline + saline i.p.; ^ *p* < 0.05, ^^ *p* < 0.01, ^^^ *p* < 0.001 vs. Yoked saline + cocaine i.p.; ^#^
*p* < 0.05, ^##^
*p* < 0.01, ^###^
*p* < 0.001 vs. Yoked cocaine + cocaine i.p. A description of the abbreviations has been placed under Figure 2.

**Figure 4 molecules-24-01125-f004:**
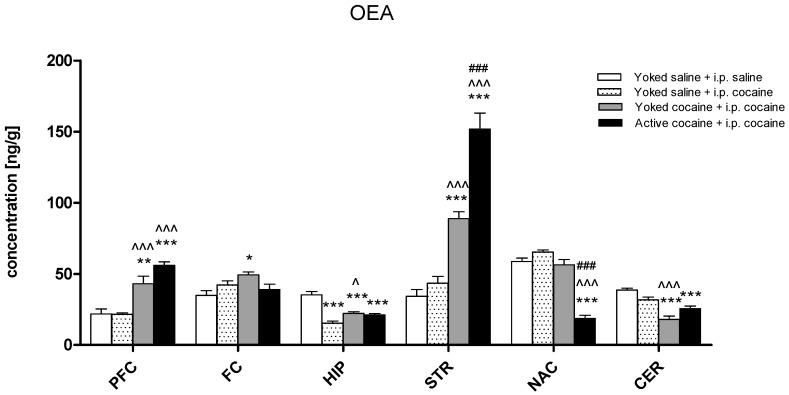
Levels of oleoylethanolamide (OEA) in brain structures in rats previously self-administered cocaine (14 days) (Active cocaine) or which passively received cocaine (Yoked cocaine) or saline (Yoked saline) and underwent extinction training (10 days) and drug-induced reinstatement (10 mg/kg cocaine i.p. or saline i.p.). N = 7–8 rats/group. All data are expressed as the means ± SEM. * *p* < 0.05, ** *p* < 0.01, *** *p* < 0.001 vs. Yoked saline + saline i.p.; ^ *p* < 0.05, ^^ *p* < 0.01, ^^^ *p* < 0.001 vs. Yoked saline + cocaine i.p.; ^#^
*p* < 0.05, ^##^
*p* < 0.01, ^###^
*p* < 0.001 vs. Yoked cocaine + cocaine i.p. A description of the abbreviations has been placed under Figure 2.

**Figure 5 molecules-24-01125-f005:**
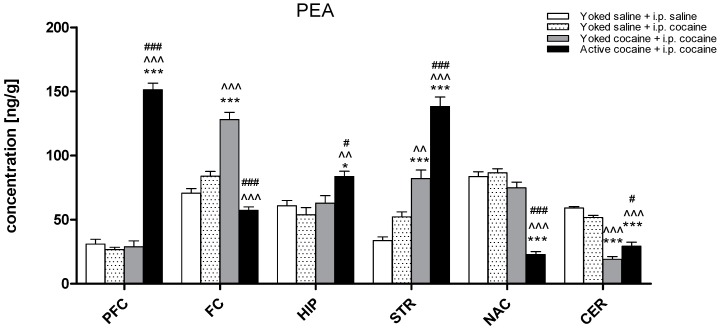
Levels of palmitoylethanolamide (PEA) in brain structures in rats previously self-administered cocaine (14 days) (Active cocaine) or which passively received cocaine (Yoked cocaine) or saline (Yoked saline) and underwent extinction training (10 days) and drug-induced reinstatement (10 mg/kg cocaine i.p. or saline i.p.). N = 7–8 rats/group. All data are expressed as the means ± SEM. * *p* < 0.05, ** *p* < 0.01, *** *p* < 0.001 vs. Yoked saline + saline i.p.; ^ *p* < 0.05, ^^ *p* < 0.01, ^^^ *p* < 0.001 vs. Yoked saline + cocaine i.p.; ^#^
*p* < 0.05, ^##^
*p* < 0.01, ^###^
*p* < 0.001 vs. Yoked cocaine + cocaine i.p. A description of the abbreviations has been placed under Figure 2.

**Figure 6 molecules-24-01125-f006:**
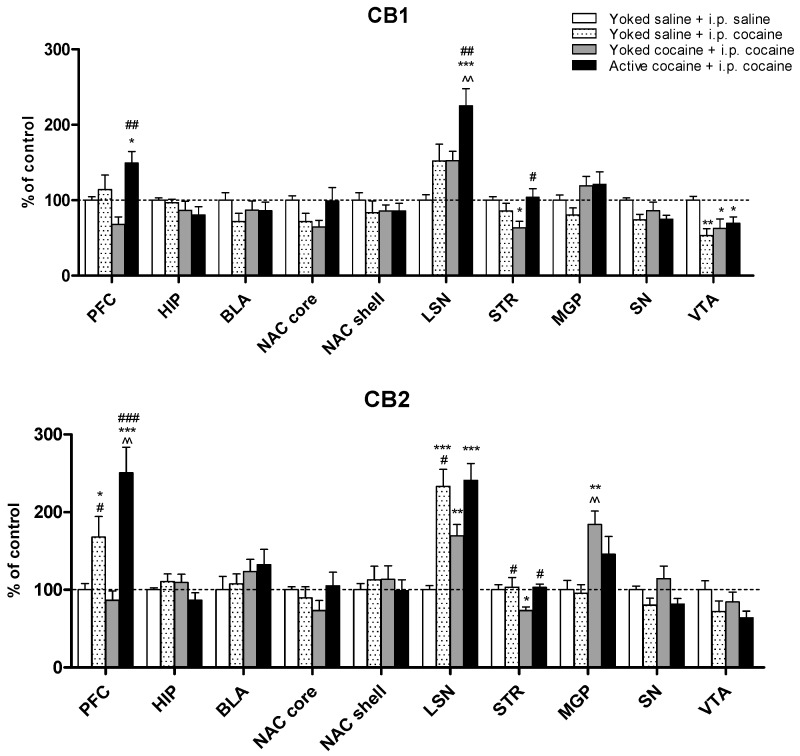
Results of immunohistochemistry analysis of membrane’s cannabinoid receptor 1 and 2 in the reinstatement cocaine phase. All data are expressed as the means ± SEM. N =  7 rats/group. * *p* < 0.05, ** *p* < 0.01, *** *p* < 0.001 vs. Yoked saline + saline i.p.; ^^ *p* < 0.01 vs. Yoked saline + cocaine i.p.; ^#^
*p* < 0.05, ^##^
*p* < 0.01 ^###^
*p* < 0.001 vs. Yoked cocaine + cocaine i.p. PFC—prefrontal cortex; HIP – hippocampus, BLA—basolateral and basomedial amygdala, NAC core or shell—nucleus accumbens core or shell, LSN—lateral septal nuclei, STR—dorsal striatum, MGP—medial globus pallidus SN—substantia nigra, VTA—ventral tegmental area.

**Figure 7 molecules-24-01125-f007:**
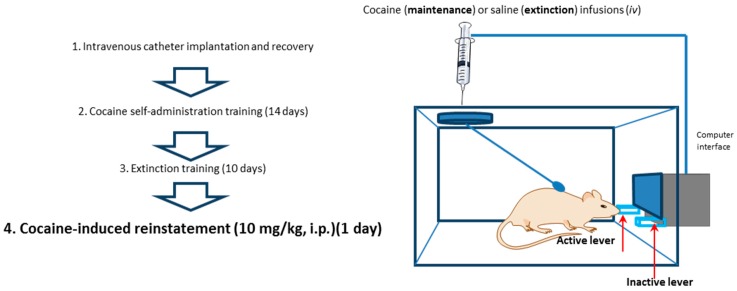
Illustration of cocaine self-administration experiments.

**Figure 8 molecules-24-01125-f008:**
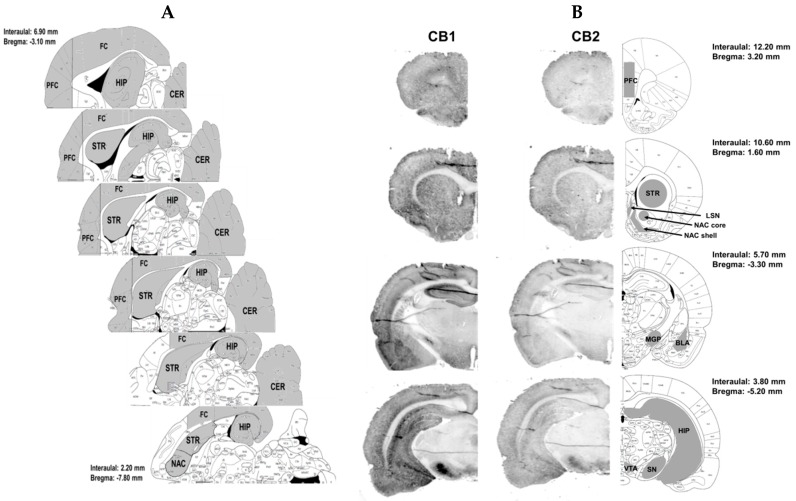
Illustration of rat brain structures for detection of level of endocannabinoid and NAEs (panel **A**; horizontal section) or immunostaining of cannabinoid CB1 and CB2 receptors after passive saline administration, extinction training, and saline injection during reinstatement (Yoked saline + i.p. saline) (panel **B**; coronal sections). BLA—basolateral and basomedial amygdala, CER—cerebellum, FC—frontal cortex, HIP—hippocampus, LSN—lateral septal nucleus, MGP—medial globus pallidus, NAC core—nucleus accumbens core, NAC shell—nucleus accumbens shell, PFC—prefrontal cortex, SN—substantia nigra, STR—dorsal striatum, and VTA—ventral tegmental area. According to the Rat Brain Atlas [34].

**Table 1 molecules-24-01125-t001:** Changes in the endocannabinoid, NAE (tissue levels), and cannabinoid receptors (CB1 and CB2) in the selected rat brain structures during cocaine-induced reinstatement.

Brain Structure	AEA			2-AG			OEA			PEA			CB1			CB2		
	A	Y	YS	A	Y	YS	A	Y	YS	A	Y	YS	A	Y	YS	A	Y	YS
PFC	↑	--	--	--	--	--	↑	↑	--	↑	--	--	↑	--	--	↑	--	↑
FC	--	--	--	↓	--	--	--	↑	--	--	↑	--	--	--	--	--	--	--
HIP	↑	--	--	↑	--	--	↓	↓	↓	↑	--	--	--	--	--	--	--	--
BLA	n/a												--	--	--	--	--	--
NAC core	↓	↓	--	↑	--	--	↓	--	--	↓	--	--	--	--	--	--	--	--
NAC shell													--	--	--	--	--	--
LSN	n/a												↑	--	--	↑	↑	↑
STR	--	--	--	--	↑	--	↑	↑	--	↑	↑	--	--	↓	--	--	↓	--
MGP	n/a												--	--	--	--	↑	--
SN													--	--	--	--	--	--
VTA													↓	↓	↓	--	--	--
CER	↓	↓	--	↓	↓	--	↓	↓	--	↓	↓	--	n/a					

A—active cocaine + i.p. cocaine group, Y—yoked cocaine + i.p. cocaine group; YS—yoked saline + i.p. cocaine. PFC—prefrontal cortex; FC—frontal cortex, HIP—hippocampus, BLA—basolateral amygdala, NAC core and shell—nucleus accumbens core and shell, LSN—lateral septal nuclei, STR—dorsal striatum, MGP—medial globus pallidus, SN—substantia nigra, VTA—ventral tegmental area, CER—cerebellum. ↑—increase, ↓—decrease, --—no changed, n/a—not analyzed.

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
