# Peer review of "Cocaine-Induced Reinstatement of Cocaine Seeking Provokes Changes in the Endocannabinoid and N-Acylethanolamine Levels in Rat Brain Structures"

_molecules, 2019, doi:10.3390/molecules24061125_

Reviewer 1 Report

The authors describe the changes in endocannabinoids and the expression of cannabinoid receptors after cocaine-induced reinstatement of cocaine seeking behaviour.

 The manuscript and study will potentially benefit from addressing the following issues:

 Major concerns:

The authors need to identify significance and rationale of the study.

 The labelling and presentation of measured parameters in the figures are unclear. Experimental conditions are not labelled properly.

 The data presented does not address potential problems resulting from signal saturation particularly for lipid quantitation and immunohistochemistry      assays.

 Positive controls and appropriate assays to measure antibody specificity should be included for immunohistochemistry assays. For immunohistochemistry experiments, necessary controls are lacking. The rationale for these experiments needs to be presented and the quality of reagents needs to be adequately characterized.

 The description of      statistical methods is insufficient. The statistical significance of the findings is unclear.

 Effects of neuronal health, cocaine-induced neurotoxicity,  inflammation and post-mortem changes in lipid and protein composition should be considered and critical parameters should be included. Measurements to determine the extent of injury should be included.

 Effects of drug treatment on blood flow and tissue oxygenation / degeneration as potential alternative mechanisms should be considered and measured.

 The rationale for only using male animals is not presented.

 Validation of drug purity and basic information on drug pharmacokinetics especially with respect to the brain are missing. This information is critical for each   set of experiments and a standardized approach would enable the reader to reproduce data and experimental conditions.

The summary data shown in figure 6 are of insufficient quality; e.g. full sections with adequate controls for specificity should be shown and quantitative analytical methods and adequate controls should be used for immunohistochemistry.

 The authors need to provide time course data to substantiate their conclusions.

 Adequate measures for assessing differential dehydration during freezing and storage should be included.

 Adequate measures for normalisation of data beyond tissue wet weight (protein controls) should be included.

 Minor concerns:

The manuscript should be proofread carefully to eliminate typographical and syntax errors. Especially the discussion section contains multiple issues that make it difficult to understand what the authors are trying to say.

The referencing is insufficient and needs to be updated with the relevant literature.

Author Response

We would like to thank the Reviewers for the assessment of the manuscript as well as for the helpful comments. We have followed all the Reviewers’ requirements.

Reviewer 2 Report

The authors of this manuscript investigated changes in endocannabinoids (AEA and 2-AG) and endogenous non-cannabinoid lipids (PEA and OEA) levels as well as changes in the expression of CB1 and CB2 receptors caused by cocaine priming. Biphasic changes in the AEA, 2AG, OEA, PEA levels as well as in the cannabinoid receptor expression were observed in different brain regions. Overall, it is an interesting study with some novel findings. My major concerns are regarding the manuscript preparations, data analysis and interpretations.

1.     The manuscript requires extensive proofreading for syntax, grammatical errors as well as editing to make the article better organized and easier to read.

2.     It is not clear why the authors decided to target particular brain regions and how the observed changes in these regions are related to cocaine-primed reinstatement. In the introduction and discussion sections the authors should elaborate on the connection between drug taking/ seeking and neuroadaptation in these particular brain regions.

3.     The entire manuscript is about changes in the endocannabinoid and NAEs levels caused by cocaine primed reinstatement in different groups of rats (exposed to response-contingent or non-contingent cocaine and primed with saline or cocaine). Yet the authors fail to present data from the reinstatement experiment for all four groups. Which groups showed reinstatement of drug seeking?

4.     A proper control group is missing. The authors need to demonstrate that reinstatement of responding occurred only in the active cocaine +ip cocaine group but not in rats primed with saline (active cocaine +ip saline group). Otherwise, it is impossible to draw a conclusion that cocaine priming elicited reinstatement of drug seeking and caused changes in the endocannabinoid and NAE systems.

5.     It is not clear why yoked control groups were used for the reinstatement experiment and brain analyses. Yoked control groups are typically used in motivated behavioral experiments to address behavioral or neuronal changes contingent upon learning (e.g. reinforcement or punishment; stimulus-outcome, response-outcome). The authors should present strong rationale for using these groups.

6.     Throughout the results sections the authors use terms such as “as compared to yoked group”, “yoked-saline group” or, “ control group”. There are 3 yoked control groups in this study; therefore, comparisons should be described more precisely (e.g. yoked saline/saline or yoked saline/cocaine, yoked cocaine/ saline).

7.     The interpretations of PEA and OEA biphasic changes are unclear. Do the authors suggest that the decreased in PEA and OEA in the nucleus accumbens is related to cocaine sensitization that occurred after cocaine priming or habit formation, or both?

8.     The authors may consider to reorganize the discussion section based on either molecular/receptor expression or brain regions.

Author Response

We would like to thank the Reviewers for the assessment of the manuscript as well as for the helpful comments. We have followed all the Reviewers’ requirements.

1. The manuscript requires extensive proofreading for syntax, grammatical errors as well as editing to make the article better organized and easier to read.
Response:
We apologize for it. The major revision of the manuscript has been made. Additionally, the manuscript has been corrected by a native English speaker.
2. It is not clear why the authors decided to target particular brain regions and how the observed changes in these regions are related to cocaine-primed reinstatement. In the introduction and discussion sections the authors should elaborate on the connection between drug taking/ seeking and neuroadaptation in these particular brain regions.
Response:
We fully agree with Reviewer’ comment and we have supplemented the missing information about the specific regions isolated and their functions (the revised version: page: 2; lines: 73-78). The detailed information about dissection of rat brain structures has been presented in the new figure (Fig. 8).

3. The entire manuscript is about changes in the endocannabinoid and NAEs levels caused by cocaine primed reinstatement in different groups of rats (exposed to response-contingent or non-contingent cocaine and primed with saline or cocaine). Yet the authors fail to present data from the reinstatement experiment for all four groups. Which groups showed reinstatement of drug seeking?

Response:
The missing information about the behavioral data showed reinstatement of drug seeking has been supplemented in the Results (the revised version: page: 3, lines: 92-95 and 103-109; Fig. 1).

4. A proper control group is missing. The authors need to demonstrate that reinstatement of responding occurred only in the active cocaine +ip cocaine group but not in rats primed with saline (active cocaine +ip saline group). Otherwise, it is impossible to draw a conclusion that cocaine priming elicited reinstatement of drug seeking and caused changes in the endocannabinoid and NAE systems.
Response:
As shown previously by us (Pomierny-Chamiolo et al. Cocaine self-administration, extinction training and drug-induced relapse change metabotropic glutamate mGlu5 receptors expression: Evidence from radioligand binding and immunohistochemistry assays. Brain Res. 2017 Jan 15;1655:66-76) and other authors (Perry et al., Escalation of i.v. cocaine self-administration and reinstatement of cocaine-seeking behavior in rats bred for high and low saccharin intake. Psychopharmacology (Berl). 2006 Jun;186(2):235-45) the saline challenge in rats previously self-administered cocaine does not reinstate active lever presses, what means that saline injection (or stress related with the procedure) does not evoke relapse. If so, the same changes in the endocannabinoid or NAE levels should be observed in all yoked controls and active cocaine rats in this study.

5. It is not clear why yoked control groups were used for the reinstatement experiment and brain analyses. Yoked control groups are typically used in motivated behavioral experiments to address behavioral or neuronal changes contingent upon learning (e.g. reinforcement or punishment; stimulus-outcome, response-outcome). The authors should present strong rationale for using these groups.
Response:
We fully agree with this important comment. Yoked controls groups in this paper underwent the same surgery and behavioral procedures (lever-press training) as cocaine self-administered rats to avoid the effect of procedure manipulation on the neurotransmitter levels.

6. Throughout the results sections the authors use terms such as “as compared to yoked group”, “yoked-saline group” or, “ control group”. There are 3 yoked control groups in this study; therefore, comparisons should be described more precisely (e.g. yoked saline/saline or yoked saline/cocaine, yoked cocaine/ saline).

Response:
As mentioned, the whole manuscript has been corrected and we have described more precisely the yoked groups.

7. The interpretations of PEA and OEA biphasic changes are unclear. Do the authors suggest that the decreased in PEA and OEA in the nucleus accumbens is related to cocaine sensitization that occurred after cocaine priming or habit formation, or both?
Response:
We have addressed this question in the Discussion (the revised version: page: 11, lines: 328-331):
“The decrease in PEA and OEA in nucleus accumbens observed in our study may confirm the increased cocaine sensitization as a result of the habit forming, since the same changes have not been observed in cocaine-experienced animals following repeated passive (yoked) treatment.”

8. The authors may consider to reorganize the discussion section based on either molecular/receptor expression or brain regions.
Response:
The previous version of discussion section has been organized in two major parts:
1) changes in the endocannabinoid system (levels of endocannabinoids and cannabinoid receptors) in relapse;
2) changes in the endocannabinoid-like molecules (which not bind with CB receptors) in relapse.
In our opinion, this division seems to be transparent and helpful in the interpretation of the results. Since none of the other two reviewers did not address this point, so we have decided to leave this section as it was.

Reviewer 3 Report

This manuscript describes region-specific changes in endocannabinoids and N-3 acylethanolamines levels after cocaine primed reinstatement. It identifies several changes that appear to be specific to the self-administration of cocaine and/or cocaine injection near the time of sacrifice. This manuscript is highly descriptive, and revisions are required to provide a better overview of the experimental findings. There are minor grammatical errors found throughout the manuscript. Specific critiques are listed below:

-       Please include relationship of NAEs to endocannabinoid signaling in introduction

-       Please show lever pressing data during “SA”, “extinction”, and “reinstatement” for all groups of rats

-       Methods – state if animals were water restricted during water training

-       Please use images from an atlas to illustrate brain regions isolated for tissue analysis

-       Please include immunohistochemistry images of both CB1 and CB2 immunoreactivity (including depiction of regions of interest) in the manuscript

-       In the conclusion, “…control cocaine reward…” should be changed to a more conservative statement

-       Please include a summary table or diagram of changes unique to cocaine self-administering animals relative to other groups to help consolidate experimental findings.

-       The behavioral data (Fig 1) notes that there are n=8/group, but the LC/MS-based data denotes n=7-8/group, and the immunohistochemistry (IHC) data denotes n=7/group. Please show behavioral data for each of the two experiments (that which used LC/MS analysis and that which used IHC analysis.

-       Line 415, 10% what?

-       State how many sections were used for IHC analysis of each brain region

Author Response

We would like to thank the Reviewers for the assessment of the manuscript as well as for the helpful comments. We have followed all the Reviewers’ requirements.

1. Please include relationship of NAEs to endocannabinoid signaling in introduction
Response:
In the revised version of the Introduction we have addressed the Reviewer’s remark (the revised version: page: 2, lines: 54-60):
“Apart from endocannabinoids, which are direct agonists of CB receptors, there are also other lipid-based mediators – NAEs (oleoylethanolamide (OEA) and palmitoylethanolamide (PEA)), and they act indirectly on the cannabinoid receptors. NAEs are the endogenous ligands of the nuclear receptor peroxisome proliferator-activated receptors (PPAR-α) and the transient receptor potential cation channel subfamily V member 1 (TRPV1) (Melis et al., 2008). PEA and OEA levels could strengthened the effect of AEA on cannabinoid receptor or TRPV1 ("entourage effect") (De Petrocellis et al., 2001) and – in consequence ˗ potentiate the effect of endocannabinoids.”

2. Please show lever pressing data during “SA”, “extinction”, and “reinstatement” for all groups of rats
Response:
The missing information about the behavioral data has been supplemented in the Results (the revised version: page: 3, lines: 92-95 and 103-109; Fig. 1).

3. Methods – state if animals were water restricted during water training
Response:
The Reviewer’s remark has been addressed in the Materials and methods section (the revised version: page: 12, lines: 359-360):
“After a week of habituation to the animal facility, animals were water-deprived for 18 h and subsequently trained to press the lever of standard operant conditioning chambers (Med-Associates, USA) under an increasing schedule of water reinforcement at a fixed ratio (FR) of 1 to 5.”
4. Please use images from an atlas to illustrate brain regions isolated for tissue analysis

Response:
The detailed information about dissection of rat brain structures has been presented in the new figure (Fig 8) and in the Introduction (the revised version: page: 2; lines: 73-78). We have also supplemented the missing information about the function of the brain structures isolated.

5. Please include immunohistochemistry images of both CB1 and CB2 immunoreactivity (including depiction of regions of interest) in the manuscript
Response:
As requested, the missing images have been supplemented in the Materials and methods section (Fig.8).

6. In the conclusion, “…control cocaine reward…” should be changed to a more conservative statement
Response:
According to the Reviewer’s comment, we have changed the indicated phrase into more precise way “influence cocaine self-administration” (the revised version: page: 16, line: 517).

7. Please include a summary table or diagram of changes unique to cocaine self-administering animals relative to other groups to help consolidate experimental findings.
Response:
According to the Reviewer’s comment, we have added the table in the Results section (the revised version: page: 8, lines: 237-238 and page: 9, lines: 249-255; Table 1).

8. The behavioral data (Fig 1) notes that there are n=8/group, but the LC/MS-based data denotes n=7-8/group, and the immunohistochemistry (IHC) data denotes n=7/group. Please show behavioral data for each of the two experiments (that which used LC/MS analysis and that which used IHC analysis.
Response:
The missing information about the behavioral data for each of the two experiments has been supplemented in the Results (the revised version: page: 4, lines: 117 ; Fig.1; page: 5, lines: 139 ; Fig.2; page: 6, lines: 161, Fig.3; page: 7, lines: 186, Fig.4; page: 8, lines: 205, Fig.5).
9. Line 415, 10% what?
Response:
We have addressed the Reviewer’s remark (“10% sucrose”) (the revised version: page: 14, line: 468).
10. State how many sections were used for IHC analysis of each brain region
Response:
For IHC analysis only one (the same) section from each brain region has been studied according the Rat Brain Atlas (Paxinos). We have supplemented a figure presented the brain areas to study the expression of CB1 and CB2 receptors and added more information to the Materials in Methods section (the revised version: page: 2, line: 73-78 ; Fig. 8, panel B).

Round  2

Reviewer 2 Report

The authors have addressed my commons appropriately. I have no more question. 

Author Response

1. English language and style are fine/minor spell check required .

Response:

The manuscript has been corrected by a native English speaker.

Reviewer 3 Report

The manuscript has been greatly improved. 

While the authors showed validation data for the CB1 antibody, there is no such data for the CB2 antibody. These antibodies are notoriously non-specific, so this is a critical concern with the data based on CB2 immunohistochemistry. Please provide compelling validation data for the CB2 antibody or remove data. 

Author Response

1. English language and style are fine/minor spell check required.

Response:

The manuscript has been corrected by a native English speaker.

2. While the authors showed validation data for the CB1 antibody, there is no such data for the CB2 antibody. These antibodies are notoriously non-specific, so this is a critical concern with the data based on CB2 immunohistochemistry. Please provide compelling validation data for the CB2 antibody or remove data.

Response:

Anti-CB2 antibody (Santa-Cruz Biotechnology, sc-10076) has been previously validated (Svízenská et al., 2013), and this information has been added to the Materials and Methods (the revised version: page: 14, lines: 478). This antibody is the same as used in several researches (Fernández-Trapero et al., 2017; Li and Kim et al., 2015; Lopez-Rodriguez et al., 2015; Espejo-Porras et al., 2015; Svízenská et al., 2013; Rubio-Araiz et al., 2008; Raborn et al., 2008; Romero-Sandoval et al., 2008; Molina-Holgado et al., 2007), and published previously in the impacted journals, i.e. Disease Models & Mechanisms; Neuroscience; PLoS ONE; Journal of Neuroimmune Pharmacology; Journal of Histochemistry and Cytochemistry; Molecular and Cellular Neurosciences; Anesthesiology; European Journal of Neuroscience.